# Low-Glycemic Load Diets and Thyroid Function: A Narrative Review and Future Perspectives

**DOI:** 10.3390/nu16030347

**Published:** 2024-01-25

**Authors:** Ioanna Partsalaki, Georgios K. Markantes, Marina A. Michalaki

**Affiliations:** 1Department of Nutrition and Dietetics, University Hospital of Patras, 26504 Rio, Greece; ioannapartsalaki@gmail.com; 2Division of Endocrinology, Department of Internal Medicine, School of Health Sciences, University of Patras, 26504 Rio, Greece; markantes@upatras.gr

**Keywords:** low-glycemic load diets, glycemic load, glycemic index, thyroid function, thyroid hormones, thyroid cancer

## Abstract

Nutrition and calorie intake are associated with subtle changes of thyroid function tests in subjects with an intact Hypothalamic-Pituitary-Thyroid axis. Iodine deficiency and extreme fluctuations in calorie intake, such as those that occur during periods of starvation or overfeeding could lead to alterations in thyroid hormones. The dietary macronutrient and micronutrient composition could also influence the thyroid function. Recently, Low-Glycemic Load (LGL) diets have become very popular and are effective in the treatment and/or prevention of several medical conditions, including diabetes, obesity, cardiovascular disease, and epilepsy. In this review, we report on the available data from the literature regarding the association between LGL diets and thyroid function or dysfunction. Several studies conducted in this field to date have yielded inconsistent results.

## 1. Introduction

Overt thyroid dysfunction, such as hyperthyroidism and hypothyroidism, results in significant alterations in thyroid hormones that unequivocally affect body weight. However, individuals with an intact hypothalamic-pituitary-thyroid axis (HPT) may experience fluctuations in thyroid function tests that persist within the normal reference range under specific circumstances, with uncertain clinical significance [1]. It appears that nutritional factors [2,3,4,5,6,7,8,9,10] and extreme fluctuations in calorie intake, such as those encountered during periods of starvation or overfeeding [11,12,13,14,15], could influence thyroid function tests. Furthermore, in euthyroid subjects, obesity can lead to subtle yet definite alterations in thyroid function, with an unclear impact on body weight [16,17,18,19]. These observations suggest that dietary patterns may influence thyroid function in individuals without HPT axis pathology. Recently, Low-Glycemic Load (LGL) diets have gained popularity and shown promise in the prevention and/or treatment of various medical conditions, including diabetes, obesity, cardiovascular disease, and epilepsy. In this review, we aim to explore the challenges associated with nutritional management using LGL diets and their impact on thyroid function, both in health and disease.

## 2. Thyroid Hormones Physiology

Thyroid hormones (THs) encompass 3,3′,5-triiodo-L-thyronine (T3) and thyroxine (T4) [20]. While circulating T4 is exclusively synthesized and released by the thyroid gland, approximately 20% of circulating T3 originates from the thyroid gland, with the remaining 80% resulting from the deiodination of T4 in the peripheral tissues, via deiodinase (DIO) enzymes [21]. These enzymes—namely DIO1, DIO2, and DIO3—play a crucial role in modulating circulating and intracellular THs concentrations. Possessing a selenocysteine group, they catalyze the deiodination of T4 by removing an iodine atom, from either its outer or inner ring. The outer ring deiodination of T4 results in the active T3, while the deiodination of the inner ring leads to the inactive reverse triiodothyronine (rT3) form [22]. Further deiodination of T3 and rT3 generates the metabolites 3,5 (3,5–T2) and 3,3′-diiodothyronine (3,3′–T2), which are active and inactive, respectively [23]. Moreover, the metabolism of THs involves processes such as deamination and decarboxylation, contributing to the degradation of THs. Deaminated acetic acid T4 metabolites include 3,3′,5,5′-tetraiodothyroacetic acid (Tetrac), which acts as an antagonist for the integrin receptor ανβ3, and 3,3′,5-triiodothyroacetic acid (Triac), which serves as a ligand for the thyroid hormone receptor (TR). Additionally, during sepsis, diiodothyronine (DIT) and possibly also monoiodothyronine (MIT) are formed through the oxidative degradation of THs in activated macrophages or leukocytes. Thus, there exists a broad spectrum of TH metabolites collectively known as the ‘thyronome’, with some exhibiting thyromimetic actions [23]. More than 99% of circulating THs are bound to specific binding proteins: the thyroxine-binding globulin (TBG), transthyretin (TTR), and albumin. Unbound THs enter their target cells through specific membrane transporters, such as the monocarboxylate (MCT) family and organic anion transporters (OATPs) [24]. T3 exerts its effects via the TRs, which are located in the nucleus and bind to DNA. These receptors are distributed widely throughout the body’s tissues, enabling T3 to regulate numerous gene expressions in a genomic manner. There are two types of TRs, the alpha (α) and beta (β) forms, encoded by separate genes, localized in human chromosomes 17 and 3, respectively [25]. The mRNA of TRα and TRβ undergoes alternative splicing, giving rise to three main isoforms: TRα1, TRα2, and TRβ1, which have differential expression in various tissues. TRα is the predominant form in the heart, skeleton, and brain, whereas TRβ is predominant in the liver and pituitary. In infants and children, THs play a crucial role in brain development and overall growth. In adults, THs significantly influence skeletal integrity, cardiovascular function, and metabolism. Notably, THs increase the basal metabolic rate and thermogenesis [26] and regulate carbohydrate [27] and lipid homeostasis [28]. Additionally, the non-genomic actions of THs, mediated by T4 binding to cell membrane receptors, have been elucidated [24]. In recent decades, thyroid hormone analogues have been designed to selectively mimic the effects of thyroid hormones by binding to either TRα or TRβ receptors; they are utilized in euthyroid subjects to harness beneficial effects while minimizing potential harm in other tissues [29]. For instance, numerous trials have explored the positive impact of TRβ analogs in reducing Low Density Lipoprotein (LDL) cholesterol without adverse effects on the heart and skeleton, as seen in eprotirome [30]. Additionally, TRβ analogues have shown promise in reducing body weight by increasing the basal metabolic rate [31]. Nevertheless, these compounds are not currently recommended in clinical practice due to uncertainties surrounding both their benefits and potential side effects [29]. Furthermore, thyroid analogues, like the naturally occurring T3 metabolite Triac or diiodothyropropionic acid (DITPA), have been employed in the treatment of the rare genetic syndromes associated with thyroid hormone resistance, or Allan-Herndon-Dudley syndrome. Their utility relies in their increased binding affinity to TRs, as well as their ability to enter the brain neurons without relying on the Monocarboxylate transporter 8 (MCT8) [32,33]. The production of THs is regulated by the HPT axis. Hypothalamic thyrotropin-releasing hormone (TRH) stimulates the synthesis and release of pituitary thyrotropin (thyroid-stimulating hormone, TSH), which, acting through its receptor (TSHR) on follicular thyroid cells, initiates and facilitates all stages of THs biosynthesis and secretion. Subsequently, THs enter the brain and inhibit the synthesis and secretion of TRH and TSH to maintain their concentrations within the normal range. Consequently, primary hypothyroidism is characterized by decreased T3 and T4 and increased TSH, while primary hyperthyroidism is characterized by suppressed TSH and increased T3 and T4. Additionally, various neural, humoral, and local factors modulate the HPT axis and, under specific circumstances, contribute to modifications in the axis’s physiological function [1]. Excessive or deficient production of THs by the thyroid gland results in hyperthyroidism or hypothyroidism, respectively. The main causes of hyperthyroidism include Graves’ disease or toxic adenomas, while hypothyroidism can result from Hashimoto’s thyroiditis (HT) or thyroidectomy. Both conditions can be categorized into subclinical forms, characterized by changes in TSH and thyroid hormones (THs) within the normal range, or overt forms, which involve alterations in both TSH and THs. It is estimated that in the general population, the prevalence of thyroid function alterations is ~0.5–4% in areas with sufficient dietary iodine intake [34].

## 3. Methodology

In this narrative review, a comprehensive search of the PubMed^®^, Web of Science^®^, and Google Scholar^®^ databases was conducted between July and December 2023. Additionally, a manual search was performed to identify articles not available in these databases. The search terms employed were: “Glycemic Index”, “Glycemic Load”, “Glycemic Load Diet and Thyroid”, and “Glycemic Index and Thyroid”. The scope of our investigation encompassed studies, review papers, systematic reviews, and meta-analyses published within the past 10 years leading up to the search date. Only articles published in English were considered for inclusion in our study. Considering the limited findings, the search was expanded to include the relevant literature from the preceding years.

## 4. Effects of Nutrition, Starvation/Overfeeding, and Obesity on Thyroid Function in Euthyroid Subjects

### 4.1. Nutrition

Nutrition and calorie intake are inextricably linked with thyroid function. Adequate thyroid hormone production, in subjects with intact HPT axis, depends on sufficient dietary iodine intake, as well as on a multitude of other internal and environmental factors, such as cold exposure. THs are iodinated products of the amino acid tyrosine; therefore, iodine is an essential element for their synthesis [20]. Iodine deficiency has major health consequences. During pregnancy, iodine requirements increase significantly, and according to the Systematic Review and Meta-Analysis of Candido et al., positive effects were seen from daily iodine supplementation (200 μg/d) on iodine status and maternal thyroid concentrations, especially prior to or in the early pregnancy stages [2]. In infants and children, iodine deficiency causes irreversible intellectual disability and growth disorders, while in adults, it causes goiter and other symptoms of hypothyroidism (fatigue, cold intolerance, coarse skin, hair loss, hoarse voice, edema, weight gain, memory problems, menstrual irregularity, constipation, etc.) [3]. Despite a global application of iodine supplementation programs, such as salt iodination over the last decades, iodine deficiency remains a major public health problem in a few areas [3]. On the other hand, excessive iodine intake could lead to thyroid dysfunction, manifesting as either hyperthyroidism or hypothyroidism, particularly in individuals with pre-existing subclinical thyroid disorders. This underscores the importance of maintaining the appropriate iodine intake [4]. Additionally, among various nutritional factors, certain trace elements, such as selenium (Se), zinc (Zn), and iron (Fe), play a significant role in influencing thyroid function, and their deficiency can critically impact TH homeostasis. Notably, obese individuals have been observed to exhibit lower circulatory levels of Zn and Se, along with abnormal thyroid function. In a recent systematic review by Zavros et al., encompassing 13 randomized controlled trials, it was found that Zn supplementation affects thyroid function by increasing free triiodothyronine (FT3) levels. However, the effects of selenium were deemed controversial [5]. On the contrary, in patients with HT, Se supplementation reduced serum thyroid peroxidase antibodies (TPOAb) and thyroglobulin antibodies (TgAb) concentrations after 6 months of treatment [6]. Concerning iron (Fe) deficiency, recognized as the most prevalent nutritional deficiency worldwide, the data indicate that it may adversely affect thyroid function and autoimmunity. This impact is particularly noteworthy in pregnant women and women of childbearing age [7]. It is noteworthy that the dietary intake of minerals might be related to the overall quality of a person’s diet, as such elements are usually abundant in foods such as fruit, seeds, and vegetables.

An important factor which has raised concerns over the recent decades is that of the consumption of goitrogenic foods, which are those foods that contain substances such as goitrin, thiocyanates, and isothiocyanates, and that seem to interfere with iodine metabolism leading to hypothyroidism, mostly in animal and in vitro studies [8]. The randomized trial reported by Chartoumpekis et al. found no impact of the consumption of cruciferous vegetables on THs levels or on thyroid autoimmunity [35]. Also, one small study conducted by Kim et al. on five healthy participants showed transient decreases by 25% of radioiodine uptake from baseline after 6-h of 15.2 oz intake of kale juice, twice per day for 7 days, and no effect on TSH and FT4, despite increases in serum and urinary thiocyanate concentrations [36]. This evidence suggests that the ordinary consumption of cruciferous vegetables is unlikely to cause hypothyroidism or autoimmune thyroid disorders in humans [37]. Other toxic nutritional elements—such as lead, cadmium, chromium, nickel, aluminum, and even certain food additives, i.e., nitrates used for fish and meat preservation—may provoke or support hypothyroidism when their levels in the circulation are in excess [9].

Data regarding the effects of macronutrient composition in normocaloric diets on thyroid function in healthy individuals are scarce. In a large cross-sectional study in 4585 lean or overweight self-reported euthyroid individuals, it was found that the dietary pattern is related with subtle differences in thyroid function tests in euthyroid subjects. The authors examined 18 different food consumption patterns and plasma FT3, FT4, and TSH levels using a food frequency questionnaire [10]. They found that the frequent consumption of foods with high GI, such as fruit juices and refined bread, pasta, and rice, had a negative association with TSH concentration, and a positive association with FT3 and FT4 concentrations, while a high consumption of protein or foods rich in saturated fatty acids had a negative association with FT3 and FT4 concentrations. However, there are several limitations in this study. First, although they selected individuals without thyroid disease or taking any medication affecting thyroid function, up to 719 subjects had either THs or TSH levels outside the reference range and were excluded from analysis. Also, they did not measure thyroid autoantibodies and they did not perform thyroid ultrasonography [10] (Table 1). In another study, the same group of authors examined the relation of dietary habits (assessed by a food frequency questionnaire) with the presence or absence of thyroid autoantibodies. They found that frequent consumption of animal fats and butter was associated with positive TPOAb and/or TgAb, while frequent consumption of vegetables and of dried fruit/nuts/muesli was associated with negative thyroid autoantibodies [38].

### 4.2. Starvation/Overfeeding

In addition, it seems that thyroid function is modulated by starvation, as calorie deprivation has been shown to exert profound effects on thyroid function in individuals with intact HPA axis. In 1975, Vagenakis et al. first described that caloric deprivation for 4 weeks in nine obese subjects resulted in a striking decrease in serum T3 and an increase in rT3, which returned to baseline values during a 5-day period of refeeding [11]. These findings were confirmed later in numerous studies [12,13] and are considered as a type of non-thyroidal illness syndrome. It seems that carbohydrate deprivation, instead of starvation per se, is responsible for the T3 drop [14]. Furthermore, long-term calorie restriction induces similar alterations in THs in euthyroid individuals. In one study, 28 healthy subjects adhering to a calorie restriction diet (comprising a balanced array of foods providing over 100% of the recommended daily intake for essential nutrients and an energy intake of 1779 ± 355 kcal/day) for an average duration of 6 years were examined. The TH levels were measured and compared with those of 28 sedentary individuals matched for age and sex, as well as 28 exercising individuals matched for body fat, all of whom followed Western diets. The findings revealed that subjects undergoing long-term calorie restriction exhibited lower serum triiodothyronine (T3) values [15]. It is known that weight regain following calorie restriction is associated with a preferential recovering of fat rather than lean mass. This catch-up fat recovery is believed to result, at least in part, from suppressed skeletal muscle thermogenesis. Studies in rats have shown that adaptive thermogenesis during weight/fat regain is characterized by peripheral tissue resistance to THs, which normally promote thermogenesis and energy expenditure. In particular, starvation induces a higher expression of the THs-inactivating DIO3 in skeletal muscle and the liver, and a lower expression of the THs-activating DIO2 in skeletal muscle and DIO1 in the liver; these changes persist upon refeeding/weight regain [43,44,45]. Moreover, short-term overfeeding leads to THs alterations. A recent study by Basolo et al. examined the effects of overfeeding (200% of energy requirements) in 58 euthyroid non-obese subjects using a normal glucose tolerance test [13] (Table 1). In this study, participants underwent 24-h interventions featuring five overfeeding diets with varying macronutrient compositions in a crossover design. The diets included a low-protein overfeeding diet (LPF: 3% protein, 51% carbohydrate, and 46% fat), a high-protein overfeeding diet (HPF: 30% protein, 44% fat, and 26% carbohydrate, *n* = 51), and three overfeeding diets with normal protein content (20%): standard (50% carbohydrate and 30% fat), high-carbohydrate (75% carbohydrate and 5% fat), and high-fat (60% fat and 20% carbohydrate). The changes in thyroid hormones (THs) observed after the overfeeding diets were dependent on the macronutrient composition. Following HPF, plasma TSH decreased by 9%, rT3 remained unchanged, while mean plasma free T4 (fT4) and free T3 (fT3) slightly (4–6%) decreased. Following LPF, the mean TSH decreased by 16% and fT3 increased by 6%, while fT4 and rT3 did not change significantly. Notably, overfeeding with normal protein diets exhibited no discernible effects on THs [13].

### 4.3. Obesity

Subtle changes in thyroid function have been also described in obesity. Indeed, obese children and adults had either serum TSH, free T3 (FT3), and/or free T4 (FT4) to the upper normal range [16,17], or only a slight increase in TSH concentration without alterations in FT3 and FT4 levels [18,19]. It remains unclear whether obesity per se causes these alterations or if a type of subclinical hypothyroidism predisposes one to obesity. The first hypothesis is more likely as weight loss restores thyroid function [46]. High serum leptin levels in obese individuals, produced by adipose cells [47], could be a candidate mechanism for explaining the elevated TSH levels in obesity, as it is known that leptin stimulates TSH release [48]. Other mechanisms have also been proposed, such as the presence of resistance in TH action [49], insulin resistance, and chronic low-grade inflammation [50].

## 5. Low Glycemic Load (LGL) Diets

In 1981, Jenkins et al. demonstrated the different effects of foods on the blood glucose level and introduced the term of Glycemic Index (GI), which represents a physiological assessment of a food’s carbohydrate content through its effect on postprandial blood glucose concentrations [51]. At present, there is consensus that the glycemic response to a food or a diet is better predicted if not only the quality, but also the quantity, of carbohydrates is considered. The relevant index that combines the GI of a food with the amount of its carbohydrate content is the Glycemic Load (GL = GI multiplied by the amount of carbohydrate in a typical serving), which allows the glycemic effect of a food, a meal, or a whole diet to be compared as it is realistically consumed [52]. However, there is still confusion over the meaning of the GI and GL and the appropriate way in which they should be utilized. To facilitate their widespread use, revised versions of international tables of GI and GL values have been published [53].

Regarding diabetes mellitus pathophysiology, it has been suggested that postprandial hyperglycemia may be a contributing factor in driving hyperinsulinemia and insulin resistance [54]. Randomized controlled trials have shown that low-GI/GL diets attenuate postprandial glucose concentrations, thereby improving long term glycemic control, other established cardiometabolic risk factors and adiposity [55]; thus, low GI/GL diets may reduce the risk of certain chronic diseases [56,57]. In line with these observations is a recent umbrella review of meta-analyses of prospective cohort studies, in which it has been shown that there was a positive association between dietary GI and the risk of type 2 diabetes, coronary heart disease, and colorectal, breast, and bladder cancers, as well as between GL and the risk of type 2 diabetes, coronary heart disease, and stroke. Regarding other types of cancer, there was no significant association [58].

A few years ago, LGL diets appeared to be a promising alternative to conventional diets in the treatment and prevention of obesity [59]. However, the latest meta-analyses in both children and adults suggest no clear benefit of the LGI/LGL diet over other high GI or GL diets. According to a 2023 systematic review and meta-analysis, LGI and LGL diets do not seem to be associated with changes in adiposity, cardiometabolic, or glucometabolic markers in children with overweight or obesity [60]. In line with these results, a very recent systematic review of 10 RCTs indicated little to no difference in the change of body weight with low GI/GL diets versus higher GI/GL diets or any other diet in adults, suggesting the need for more well-designed and adequately powered studies [61].

## 6. LGL Diets Lead to Subtle Changes of Thyroid Function Tests in Euthyroid Subjects

Thyroid function parameters in euthyroid individuals undergoing weight loss through dietary interventions have only been explored in a limited number of studies. Moreover, these investigations have primarily focused on weight loss in obese subjects with or without diabetes mellitus and other co-morbidities. Several types of diets have been used, but elucidating the effect of each type of diet on thyroid function is challenging due to the simultaneous impact of the underlying disease, calorie restriction, and weight loss on the results (Table 1). Recently, LGL diets have gained popularity due to their effectiveness in promoting weight loss and improving glycemic regulation in individuals with obesity or diabetes [62]. In one study, seven healthy non-obese euthyroid young adults were randomly assigned to follow either a high-fat, low-carbohydrate diet for 7 days, followed by an isocaloric high-protein, low-carbohydrate diet for another 7 days, or the reverse sequence [39] (Table 1). The thyroid hormone levels were measured after each dietary phase. Despite the diets being designed to maintain body weight, both groups experienced a small but significant weight loss after each diet. The serum levels of thyroid-stimulating hormone (TSH) and triiodothyronine (T3) decreased after each diet, while thyroxine (T4) and reverse triiodothyronine (rT3) remained stable. This study showed that short-term LGL diets induced weight loss and decreased TSH and T3, and that the decline in T3 was more pronounced after the high-fat diet [39]. A pilot study with an excellent design has recently been published [40]; it is a randomized-controlled-crossover trial in 11 young healthy lean adults. The participants were randomized to follow two isocaloric diets for a minimum of 3 weeks, each with 1 week washout period in between; either a ketogenic diet (KD: 15% carbohydrate, 60% fat, 25% protein) and then the high-carbohydrate low-fat (HCLF: 55% carbohydrate, 20% fat, 25%protein) diet or vice versa. KD is a very low-carbohydrate diet and, consequently, it could be considered as a LGL diet, though it also causes metabolic acidosis. Both interventions led to substantial weight loss, with KD exhibiting a more pronounced effect, despite the diets being isocaloric and comparable in terms of the resting metabolic rate and physical activity levels. Their findings suggest that the influence of dietary macronutrient composition on body weight defies a straightforward explanation through the principles of thermodynamic energy balance. Notably, the thyroid hormones (THs) remained unchanged on a high-carbohydrate, low-fat (HCLF) diet, while on a ketogenic diet (KD), the serum T3 levels decreased and T4 levels increased, with the TSH levels remaining constant. The observed lower serum T3 levels on the KD might be anticipated to yield effects that are contrary to those that were observed. The authors propose that unmeasured parameters regulating the T3 action could be implicated. For instance, during KD, alterations in the intracellular T3 concentration, affinity with thyroid hormone receptors (TR), or the involvement of nuclear co-repressors may play a significant role [40] (Table 1).

## 7. LGL Diets and Weight Reduction in Hypothyroidism

It is well-established that T3, the active form of THs, regulates the basal metabolic rate, thermogenesis, and food intake [63]. An inadequate production of THs in overt hypothyroidism (either autoimmune or due to other causes) is frequently associated with weight gain, whereas thyrotoxicosis of any cause leads to weight loss [64]. The meta-analysis of Song et al., including 22 studies, showed that obesity was significantly associated with an increased risk of hypothyroidism (RR = 1.86, 95% CI 1.63–2.11, *p* < 0.001) [65]. Further analysis also revealed that obesity was clearly associated with Hashimoto Thyroiditis (HT) (RR = 1.91, 95% CI 1.10–3.32, *p* = 0.022) and with high levels of anti-TPO antibodies (RR = 1.93, 95% CI 1.31–2.85, *p* = 0.001), but not with Graves’ disease. These data also indicate the bidirectional relationship between obesity and the risk of hypothyroidism [65]. In patients with HT, treatment with L-thyroxine has been shown to induce significant decreases in body weight and BMI after euthyroidism is restored [66]. However, other studies have found that even after the restoration of euthyroidism, hypothyroid patients could experience difficulties in achieving weight loss, implying that the prevention of obesity is crucial for thyroid disorders [65]. In addition, it is known that in patients receiving adequate substitution therapy with L-thyroxine, T3 levels may frequently be in the low-normal range, despite normal FT4 and TSH levels; this may explain why these patients might struggle with weight loss, as decreased T3 levels have been linked to difficulty in maintaining or further decreasing weight [67].

Several studies have suggested that eating habits could affect the risk of manifesting inflammatory and immune diseases, including autoimmune thyroiditis (HT) [68]. Furthermore, different nutritional interventions have been tested in patients with HT. Lactose-restricted, gluten-free, Paleolithic-style, and “autoimmune protocol” diets implemented in HT patients for periods ranging from a few weeks to several months have been shown to produce significant decreases in thyroid autoantibodies and, in some instances, in TSH as well [68].

We found only one report from Esposito et al., in which the effects οf a LGL diet on 180 patients with HT was studied (Table 1) [67]. Among these patients, 108 adhered to a LGL diet (12–15% of the total daily energy intake) devoid of bread, pasta, fruit, and rice and free of goitrogenic food (experimental Group, *n* = 108), and 72 were on a low caloric diet without food restrictions (control Group, *n* = 72). The intervention period spanned 3 weeks, during which baseline and concluding assessments were made for thyroid hormones and thyroid autoantibodies. Notably, the authors did not specify whether patients with HT were undergoing levothyroxine treatment, but their thyroid function tests consistently indicated values within the normal range. Patients in both groups achieved similar body weight loss, but body fat reduction was greater in patients on the LGL diet. The thyroid hormones did not change in any diet. Intriguingly, the thyroid autoantibody levels demonstrated a marked decrease, ranging between −40% and −57%, in the experimental group, while conversely increasing in the control group. These findings underscore the potential efficacy of the LGL diet in mitigating thyroid autoantibody levels among patients with HT [41].

## 8. Impact of Low-Glycemic Load Diets on the Risk of Thyroid Cancer

Papillary thyroid cancer constitutes the predominant form of thyroid cancer, accounting for 84% of all cases [69]. Over the past three decades, there has been a consistent rise in the incidence of papillary thyroid cancer, potentially linked to the overdiagnosis of indolent microcarcinomas, a result of the widespread utilization of thyroid ultrasonography [70]. However, the possibility of a genuine rise in thyroid cancer cannot be dismissed, as larger tumors continue to be diagnosed [71]. Radiation exposure during childhood is a well-established risk factor, as evidenced by a comprehensive pooled analysis of 12 studies [72]. Additionally, emerging evidence suggests a potential association between obesity and thyroid cancer development [73]. Healthy diet, avoiding red meats and sugar, protects from cancer development [74]. Regarding thyroid cancer in a case-control study by Randi et al., the researchers explored the influence of Glycemic Load (GL) and Glycemic Index (GI) in the diet [42] (Table 1). The study included 399 subjects with histologically confirmed thyroid carcinoma (of which 274 had papillary carcinomas) within the 2 years preceding their enrollment. Dietary habits were assessed using a food-frequency questionnaire and compared with those of 616 control subjects who did not have neoplastic disease. In this study, GI and GL were determined mainly by high dietary intakes of bread, rice, and pasta, and both were associated with increased thyroid cancer risk. In detail, the OR for the highest tertile of GI compared with the lowest one was 1.70 for papillary and 1.57 for follicular thyroid cancer. The respective ORs for GL were 2.17 for papillary and 3.33 for follicular thyroid cancer [42]. Also, the European Prospective Investigation into Cancer and Nutrition (EPIC) study, a large cohort with a high heterogeneity in dietary intake across the 10 participating countries, investigated the associations between differentiated thyroid carcinoma (TC) and the intake of energy, macronutrients, GI, and GL in almost half a million participants. This study revealed significant associations between TC and high total energy and low polyunsaturated fats (PUFA) intakes. The associations with starch and sugar intakes and GI were significantly heterogeneous across Body Mass Index (BMI) groups: there was a positive association of TC with starch and GI in participants with a BMI ≥ 25 kg/m^2^, and with sugar intake in those with BMI < 25 kg/m^2^ [75].

## 9. Perspectives and Conclusions

In summary, the available data regarding the correlation between LGL diets with either thyroid function or dysfunction, or predisposition to thyroid cancer are limited and carry inherent limitations. It seems that LGL diets might induce subtle thyroid abnormalities in euthyroid subjects and may decrease the risk of thyroid cancer development. In patients with HT, LGL diets might be more effective in reducing body fat. Further research is needed to confirm the above-mentioned potential relationships. Additionally, LGL diets should be further classified based on which macronutrient replaces the removed carbohydrate-derived calories, whether it is fats or proteins. The maintenance of THs homeostasis is highly complicated and depends on multiple parameters, such as the integrity of the HPT axis, the buffering of serum THs by binding proteins, and the THs cellular transportation, intracellular metabolism, and action. Additionally, the acknowledged nongenomic actions of T4 contribute to this intricate balance. Notably, LGL diets have demonstrated superior efficacy regarding weight loss compared to other dietary regimens. However, the precise mechanisms underpinning this advantage remain elusive. The potential involvement of THs homeostasis in this observed efficacy prompts further investigation. Comprehensive studies are warranted to explore the effectiveness of LGL diets across various circumstances, scrutinizing their correlation with every facet of THs homeostasis.

## Figures and Tables

**Table 1 nutrients-16-00347-t001:** Summary of studies linking LGL Diets with alterations in thyroid function and predisposition to thyroid cancer.

Study	Participants/Study Design	Main Intervention	Main Results
Brdar, D.; et al. [10]	4585 healthy adults - Cross sectional study.	- Dietary intake was assessed through a food frequency questionnaire, containing 58 food items.	- Consumption of high-glycemic index foods was positively associated with FT3 and FT4 levels, but negatively associated with TSH levels. - The levels of FT3 and FT4 were negatively correlated with saturated fats and proteins in foods.
Basolo, A.; et al. [13]	58 euthyroid healthy adults	- 24-h dietary interventions in a crossover design including fasting, eucaloric feeding, and 5 overfeeding diets. - Acute overfeeding diets (200% of energy requirements) included three diets with 20% protein, one diet with 3% protein (low-protein overfeeding diet) and one diet with 30% protein (high-protein overfeeding diet).	- FT4 increased and TSH and FT3 decreased after 24-h fasting. - TSH, FT4 and FT3 decreased following high-protein overfeeding diet. - Greater decreases in FT3 after high-protein overfeeding diet are associated with larger decreases in FGF21. - Following low-protein overfeeding diet, the mean FT3 increased and TSH decreased, with no change in FT4. - No changes in TH were observed after normal-protein overfeeding diets.
Ullrich, I., H.; et al. [39]	7 healthy young adults	- Group 1: Isocaloric, liquid, low-carbohydrate formula, high in fat, (HF), 55% of total calories, mainly polyunsaturated fat. - Group 2: Isocaloric, liquid, low-carbohydrate formula, high in protein (HP), 35% of total calories. - Evaluation carried out on day 8 of each diet.	- T3 declined more in HF than in HP diet. - TSH declined equally after both diets.
Iacovides, S.; et al. [40]	11 healthy, normal weight adults - Randomized crossover-controlled study - One-week washout period between two diets	For a minimum of three weeks on each, participants followed two isocaloric diets: - a HCLF diet (55% carbohydrate, 20% fat, 25% protein) and, - a KD (15% carbohydrate, 60% fat, 25% protein).	- The levels of T3 decreased more following the KD, but there was no difference after the HCLF diet. The levels remained in the normal range. - TSH and T4 remained unchanged after each diet
Esposito, T.; et al. [41]	180 patients with Hashimoto Thyroiditis	- Group 1: The diet consisted of carbohydrates 12–15%, proteins 50–60%, and lipids 25–30% avoiding goitrogenic food, eggs, legumes, dairy products, bread, pasta, fruits, and rice. - Group 2: low-calorie diet without restrictions regarding the types of foods. - The diets were implemented for 3 weeks.	- There was no significant change of FT3, FT4, and TSH. - There was a significant decreased in anti-thyroglobulin Ab and anti-thyroperoxidase, and anti-microsomal Ab levels of TPO Ab after the low-carbohydrate diet.
Randi, G.; et al. [42]	Case-control study - 399 adults with histologically confirmed thyroid carcinoma and 617 control patients with other, unrelated to thyroid, acute non-neoplastic diseases	Dietary habits were derived through a food-frequency questionnaire.	GI and GL were directly associated with risk of thyroid cancer.

Abbreviations: HF, High Fat; HP, High Protein; T3, triiodothyronine; TSH, thyroid-stimulating hormone; FT3, free T3; FT4, free T4; FGF21, fibroblast growth factor 21; TH, thyroid hormones; T4, thyroxine; HCLF diet, High-Carbohydrate, Low-Fat diet; KD, Ketogenic Diet; Ab, Antibody; GI, Glycemic Index; GL, Glycemic Load.

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
