# Peer review of "Low-Glycemic Load Diets and Thyroid Function: A Narrative Review and Future Perspectives"

_nutrients, 2024, doi:10.3390/nu16030347_

Round 1
Reviewer 1 Report
Comments and Suggestions for Authors
This paper aims to “report on the available data from the literature regarding the association between low glycemic diets and thyroid function and dysfunction’. However, in its current state it is a Hodge podge of everything thyroid with low glycemic diets not mentioned until line 225 of a 388 line paper. For example, the paper entitled “Low-glycemic Load Diets and Thyroid function, reviews rarely mentioned metabolites of thyroid hormone, zinc, selenium, iron, goitrogens, starvation, overfeeding, peripheral tissue thyroid resistance which have nothing to do with the question at hand. I agree an introduction on use of low glycemic diets for obesity and diabetes mellitus treatment seem appropriate but this is never adequately tied together. Thus the paper is hard to read, unfocused and really does not set out at all to be what it intended to be. As a reader, l learned a few updates about research on the thyroid I did not know but left with my head spinning. When I read the conclusion I could not even pinpoint how the authors were concluding that low glycemic diets might induce subtle thyroid abnormalities in euthyroid subjects and may decrease risk of thyroid cancer development because the few papers mentioning this were weak and berried amongst other unfocused words.
General Comments
· Introduction is way too long and unfocused. Info on lines 46-55, 59-66, 71-85 has absolutely nothing to do with the purpose of the review. I initially thought maybe low glycemic diet loads were going to be found to be related somehow one of more of these metabolites… The introduction needs to be focused and introduce the topic not throw in random facts. Additionally, there are no references in the first paragraph. While some of this information is considered common knowledge not all of it is . References needed on lines 25, 27, 29.
· Despite the long introduction an average reader who knows little about the thyroid would not understand much about this paper. Would be good to explain that TSH increases typically with primary hypothyroid and visa versa with hyperthyroid.
· Title does not reflect manuscript
· Authors do not tie in information from Table 1.
Minor Points
· Lines 31/32. Prevention and/or treatment
· Line 55 “of them” should be omitted
· Line 122. “mental retardation”; check whether this is still acceptable terminology
· Line 122. State the other symptoms
· Line 129-131 this is very vague about the minerals, although I guess this is not the purpose of the review but maybe minerals are tied to quality of diet
· Lines 137, 164 and throughout the manuscript; concentration not level is the proper term for concentrations of a hormone, metabolite, biomarker, etc in the blood
· Line 172, poor transition
· Line 212-214, everyone knows this; it is overwritten.
· Line272, how is this possible
Author Response
REVIEWER 1
“This paper aims to “report on the available data from the literature regarding the association between low glycemic diets and thyroid function and dysfunction’. However, in its current state it is a Hodge podge of everything thyroid with low glycemic diets not mentioned until line 225 of a 388 line paper. For example, the paper entitled “Low-glycemic Load Diets and Thyroid function, reviews rarely mentioned metabolites of thyroid hormone, zinc, selenium, iron, goitrogens, starvation, overfeeding, peripheral tissue thyroid resistance which have nothing to do with the question at hand. I agree an introduction on use of low glycemic diets for obesity and diabetes mellitus treatment seem appropriate but this is never adequately tied together. Thus the paper is hard to read, unfocused and really does not set out at all to be what it intended to be. As a reader, l learned a few updates about research on the thyroid I did not know but left with my head spinning. When I read the conclusion I could not even pinpoint how the authors were concluding that low glycemic diets might induce subtle thyroid abnormalities in euthyroid subjects and may decrease risk of thyroid cancer development because the few papers mentioning this were weak and berried amongst other unfocused words”.
General Comments
- “Introduction is way too long and unfocused. Info on lines 46-55, 59-66, 71-85 has absolutely nothing to do with the purpose of the review. I initially thought maybe low glycemic diet loads were going to be found to be related somehow one of more of these metabolites… The introduction needs to be focused and introduce the topic not throw in random facts. Additionally, there are no references in the first paragraph. While some of this information is considered common knowledge not all of it is. References needed on lines 25, 27, 29”.
Thank you for your comments. Because this manuscript is addressed to a broad audience (not exclusively endocrinologists or readers who are familiar with thyroid physiology), we have chosen to produce a rather extended introduction, to help the reader capture the full spectrum of thyroid hormone mechanisms of action and actions. The information in lines 46-55, 59-66, and 71-85 of the original submission was added following suggestions of the handling academic editor. We have also added the relevant references (see lines 25-27, 29).
- “Despite the long introduction an average reader who knows little about the thyroid would not understand much about this paper. Would be good to explain that TSH increases typically with primary hypothyroid and visa versa with hyperthyroid”.
We have made an addition according to your suggestion (see lines 91-93), thank you.
- “Title does not reflect manuscript”
The title of this manuscript has been selected in collaboration with the handling editor (to be included in the journal’s special issue on low glycemic load diets), and unfortunately we cannot modify it. Thank you.
- “Authors do not tie in information from Table 1”.
We have added a reference each time a study from Table 1 is mentioned in the text (see lines 178, 211, 280, 304), thank you.
Minor Points
- “Lines 31/32. Prevention and/or treatment”
We have changed this phrase according to your suggestion (see line 32), thank you.
- “Line 55 “of them” should be omitted”
Done, thank you.
- “Line 122. “mental retardation”; check whether this is still acceptable terminology”
We have replaced the term “mental retardation” with “intellectual disability” (see line 125). Thank you.
- “Line 122. State the other symptoms”
Done (see lines 126-128), thank you.
- “Line 129-131 this is very vague about the minerals, although I guess this is not the purpose of the review but maybe minerals are tied to quality of diet”
An addition has been made (see lines 146-148), thank you.
- “Lines 137, 164 and throughout the manuscript; concentration not level is the proper term for concentrations of a hormone, metabolite, biomarker, etc in the blood”
We have made changes in accordance with your suggestion, thank you. However, the term “level” is very frequently used in the medical literature to refer to hormone concentrations in the blood.
- “Line 172, poor transition”
The sentence has been modified (see lines 185-186), thank you.
- “Line 212-214, everyone knows this; it is overwritten”.
This part has been omitted, thank you.
- “Line272, how is this possible”
Those are the findings reported by the authors of that study. This pattern (decreased TSH and T3 and stable/normal T4) is observed in several instances, such as in fasting or in critical illness (“non-thyroidal illness syndrome”).
Reviewer 2 Report
Comments and Suggestions for Authors
This is a narrative review article that analyses the association between the LGL diet and the function or dysfunction of the thyroid gland. The paper discusses several studies conducted in this field so far, commenting on the inconsistent results that have been reported.
The manuscript is generally well-prepared, although I have a few minor comments:
Abstract: The last sentence in the abstract needs to be reformulated, for example, “Several studies conducted in this field so far have yielded inconsistent results”.
Methodology: Although the literature search is limited to the last ten years, which is essentially a short period, two of the references mentioned pertain to earlier periods: Ullrich IH et al from 1985 and Randi G et al from 2008. It is necessary to accurately describe how the literature was reviewed in the Methodology section.
4.1 Nutrition: In the last paragraph of this subsection, referring to the study by Brdar et al from 2021, the authors note that a limitation of this study is that the authors did not measure thyroid antibodies. They may not have noticed that the same group of authors published a paper in 2017 in the journal Nutrients titled “Dietary Factors Associated with Plasma Thyroid Peroxidase and Thyroglobulin Antibodies”, the results of which would be worth discussing in this review.
4.2 Starvation/overfeeding: In the part of the sentence mentioned in lines 205 and 206, 'Following LPF or HPF, plasma TSH decreased by 13-16%...' the statement is not accurate and is not consistent with the claim stated in Table 1.
References: The references are not correctly formatted; according to the journal guidelines, only the volume of the journal is provided, not the issue. Pages are missing in several cited references. The major flaw is that reference 14 is split into two references, 14 and 15, causing a displacement of all other cited references, leading to incompatibility with the text of the paper. This must be corrected, and thorough verification is required before resubmitting the manuscript.
Author Response
REVIEWER 2
“This is a narrative review article that analyses the association between the LGL diet and the function or dysfunction of the thyroid gland. The paper discusses several studies conducted in this field so far, commenting on the inconsistent results that have been reported.
The manuscript is generally well-prepared, although I have a few minor comments:
Abstract: The last sentence in the abstract needs to be reformulated, for example, “Several studies conducted in this field so far have yielded inconsistent results”.
We have changed the last sentence of the abstract according to your suggestion (see line 16), thank you.
“Methodology: Although the literature search is limited to the last ten years, which is essentially a short period, two of the references mentioned pertain to earlier periods: Ullrich IH et al from 1985 and Randi G et al from 2008. It is necessary to accurately describe how the literature was reviewed in the Methodology section”.
Thank you for your comment. In the last sentence of the Methodology section, we state that because of the limited results of our initial search (limited to the last 10 years) we expanded our search to include all the relevant studies we could find (see lines 111-112).
“4.1 Nutrition: In the last paragraph of this subsection, referring to the study by Brdar et al from 2021, the authors note that a limitation of this study is that the authors did not measure thyroid antibodies. They may not have noticed that the same group of authors published a paper in 2017 in the journal Nutrients titled “Dietary Factors Associated with Plasma Thyroid Peroxidase and Thyroglobulin Antibodies”, the results of which would be worth discussing in this review”.
Thank you for your suggestion. We have included a comment regarding the study you pointed out (see lines 178-183).
“4.2 Starvation/overfeeding: In the part of the sentence mentioned in lines 205 and 206, 'Following LPF or HPF, plasma TSH decreased by 13-16%...' the statement is not accurate and is not consistent with the claim stated in Table 1”.
Thank you for your comment. We have modified the above-mentioned part (see lines 218-221) and added information in Table 1.
“References: The references are not correctly formatted; according to the journal guidelines, only the volume of the journal is provided, not the issue. Pages are missing in several cited references. The major flaw is that reference 14 is split into two references, 14 and 15, causing a displacement of all other cited references, leading to incompatibility with the text of the paper. This must be corrected, and thorough verification is required before resubmitting the manuscript”.
Thank you for your comment. The reference list has been revised according to the journal’s instructions for authors. The error concerning the splitting of reference 14 in two has been corrected.
Reviewer 3 Report
Comments and Suggestions for Authors
The manuscript submitted for review is a comprehensive literature review on the impact of diets with varying carbohydrate compositions on thyroid-metabolic status.
Due to the prevalence of excessive body weight in populations with a Western lifestyle and the usen of various dietary approaches for weight reduction, the topic addressed by the authors is important and relevant.
I have no objections to the method of selecting publications subjected to analysis.
The paper is exhaustively and engagingly written, and it has a high likelihood of being cited.
Minor comments:
Line 11 (subtitle) – „ the effect of micronutrients" could be added
114-116 As well as a multitude of other internal and environmental factors, such as cold exposure.
158 Instead of the authors' names, the studied population is provided beforehand.
275 A concluding sentence summarizing the findings of this study is needed.
341The topic of thyroid function has been expanded to include thyroid cancer. It is worthwhile to incorporate this into the title or at least in the keywords.
Author Response
REVIEWER 3
“The manuscript submitted for review is a comprehensive literature review on the impact of diets with varying carbohydrate compositions on thyroid-metabolic status. Due to the prevalence of excessive body weight in populations with a Western lifestyle and the usen of various dietary approaches for weight reduction, the topic addressed by the authors is important and relevant. I have no objections to the method of selecting publications subjected to analysis. The paper is exhaustively and engagingly written, and it has a high likelihood of being cited”.
Thank you very much for your comments.
Minor comments:
“Line 11 (subtitle) – „ the effect of micronutrients" could be added”
An addition has been made (see line 11), thank you.
“114-116 As well as a multitude of other internal and environmental factors, such as cold exposure”.
An addition has been made (see lines 118-119), thank you.
“158 Instead of the authors' names, the studied population is provided beforehand”.
We have rephrased the sentence according to your suggestion (see lines 165-169), thank you.
“275 A concluding sentence summarizing the findings of this study is needed”.
An addition has been made (see lines 284-286), thank you.
“341The topic of thyroid function has been expanded to include thyroid cancer. It is worthwhile to incorporate this into the title or at least in the keywords”.
Since the title of this manuscript has been selected in collaboration with the handling editor (to be included in the journal’s special issue) and unfortunately we cannot modify it, we have just added the term “thyroid cancer” in the keywords (see line 18). Thank you.
Round 2
Reviewer 1 Report
Comments and Suggestions for Authors
The authors streamlined and focused this manuscript in their revision. It would be useful for those interested in low GI diets.